# Nuclear Molecular Imaging of Disease Burden and Response to Treatment for Cardiac Amyloidosis

**DOI:** 10.3390/biology11101395

**Published:** 2022-09-24

**Authors:** Min Zhao, Raffaella Calabretta, Josef Yu, Patrick Binder, Shuo Hu, Marcus Hacker, Xiang Li

**Affiliations:** 1Department of Nuclear Medicine, Xiangya Hospital, Central South University, Changsha 410008, China; 2Division of Nuclear Medicine, Department of Biomedical Imaging and Image-Guided Therapy, Vienna General Hospital, Medical University of Vienna, Währinger Gürtel 18-20, Floor 3L, 1090 Vienna, Austria; 3National Clinical Research Center of Geriatric Disorders, Xiangya Hospital, Central South University, Changsha 410008, China

**Keywords:** cardiac amyloidosis, bone scintigraphy, SPECT/CT, PET, diagnosis, therapy response

## Abstract

**Simple Summary:**

Cardiac amyloidosis (CA) is characterized by extracellular infiltration and deposition of amyloid fibrils primarily derived from the circulating transthyretin protein (TTR) or immunoglobulin light chain (AL). With the development of non-invasive diagnostic approaches and the emergence of new pharmacotherapeutic treatments for CA, the transformative effects of bone scintigraphy have been important in diagnosing TTR-CA. Positron emission tomography (PET) imaging is another promising, non-invasive option for the diagnosis of CA and may help differentiate between ATTR and AL amyloidosis. Bone-seeking single-photon emission tomography/computed tomography (SPECT/CT) quantification and amyloid-targeting PET imaging could be useful as a new strategy for disease burden and therapy monitoring to provide more insights into therapy response assessed by quantifying the amyloid burden in CA.

**Abstract:**

Cardiac amyloidosis (CA) is a heterogeneous group of diseases in which extracellular insoluble amyloid proteins are deposited in specific organs and tissues locally or systemically, thereby interfering with physiological function. Transthyretin protein (TTR) and light chain (AL) amyloidosis are the most common types of cardiac amyloidosis. Radionuclide bone scintigraphy has recently become the most common non-invasive test for the diagnosis of TTR-CA but is of limited value for the diagnosis of AL-CA. PET has proved promising for the diagnosis of CA and its applications are expected to expand in the future. This review summarizes the current bone scintigraphy and amyloid-targeting Positron emission tomography (PET) imaging, the binding imaging properties of radiotracers, and the values of diagnosis, prognosis, and monitoring therapy response in CA.

## 1. Introduction

In cardiac amyloidosis (CA), extracellular deposits of amyloid fibrils originating from circulating immunoglobulin amyloid light chains (AL) or transthyretin proteins (TTR) accumulate and cause extracellular infiltration. AL and TTR amyloidosis account for over 95% of clinical CA [1]. Other forms of systemic amyloidosis, such as amyloid A (AA) amyloidosis, affect the heart but are rare [2]. Because the phenotypic expression, clinical course, and therapy for AL and TTR amyloidosis are quite different, it is desirable to develop a more comprehensive and specific diagnostic method that is capable of recognizing and distinguishing CA phenotypes early in the disease process.

Endomyocardial biopsy (EMB) testing for amyloid deposits by Congo red staining is the gold standard for diagnosing CA. However, due to the heterogeneity of the disease, EMB may fail to provide sufficient information. In addition, as an inherently invasive procedure, in 6% of cases it causes complications such as arrhythmia, cardiac perforation, acute pericardial tamponade, accidental arterial puncture, and pneumothorax [3,4].

CA diagnosis and treatment outcomes have been developed by introducing a non-invasive imaging approach and through the emergence of new treatments. TTR-CA treatment is mainly based on stabilizing TTR and reducing its production. Although liver transplantation remains the gold standard for therapy in hereditary TTR patients, newly available pharmaceuticals have recently emerged to ameliorate TTR amyloidosis-associated outcomes. A tetramer stabilizer, tafamidis, was initially approved for TTR-CA treatment. Several other medications are under investigation. Furthermore, therapy directed at reducing the proliferating plasma cell and the circulating amyloid light chains is the most important aspect of AL-CA management. Principal strategies involve chemotherapy, autologous stem cell transplantation (ASCT), and monoclonal antibodies such as daratumumab, which are specific to the plasma cell [5].

Characteristic patterns of cardiac magnetic resonance (CMR) and echocardiography are highly suggestive but not diagnostic [6]. In current clinical practice, nuclear imaging has become the most reliable non-invasive modality available that can accurately diagnose AL from TTR amyloidosis when there is lacking evidence of a monoclonal protein. Therefore, it is becoming increasingly popular as an alternative diagnostic method for patients with suspected CA.

In this regard, the purpose of this review is to summarize and discuss the current state of nuclear molecular imaging, including bone-seeking scintigraphy and amyloid-targeting PET for the diagnosis and monitoring of prognosis and treatment in CA.

## 2. Binding Properties of Molecular Imaging in CA

Current bone-seeking radiotracers available include Technetium (^99m^Tc)-labeled pyrophosphate (PYP), 3,3-diphosphono-1,2-propanodicarboxylic acid (-DPD), and hydroxymethylene diphosphonate (HMDP). The favorable efficacy of these agents shows equivalent diagnostic accuracy for detecting TTR-CA [6,7,8], but another frequently used bone-seeking tracer (^99m^Tc-labled methylene diphosphonate (MDP)) is not recommended because it lacks sensitivity for diagnosis of TTR-CA [9]. The exact mechanism by which bone-seeking tracers bind and are retained in the heart remains not completely understood. It has been hypothesized that the radiopharmaceuticals bind to amyloid deposits, with greater microcalcifications in ATTR than those in AL [10,11]. However, calcification alone cannot completely account for the affinity of bone-seeking tracers to TTR because patients with Phe64Leu mutation-related TTR have low sensitivity for bone scintigraphy (DPD and HMDP) [12]. In contrast to SPECT tracers, which mostly bind to microcalcifications rather than directly to fibrils, at least four positron emission tomography (PET) tracers that directly bind to amyloid fibrils have been investigated: ^11^C-Pittsburgh compound B (^11^C-PiB) and ^18^F-labeled radiotracers such as ^18^F-florbetapir, ^18^F-florbetaben, and ^18^F-flutemetamol. Although amyloid PET tracers are not specifically connected to cardiac fibrils, they appear to have a greater affinity for AL than for ATTR amyloidosis, as evidenced by an autoradiographic study [13] which showed that the specific uptake of ^18^F-florbetapir was significantly increased in AL-CA samples compared to TTR-CA (2.48 ± 0.40 vs. 1.52 ± 0.22 DPM/mm^2^; *p* < 0.001; *n* = 10, respectively), indicating the different binding mechanisms and distinct biologic properties of CA disease entities (Figure 1).

## 3. Bone Scintigraphy

### 3.1. Diagnosis

TTR-CA is an under-recognized cause of heart failure in older adults. ATTR includes the variant or hereditary type (<10% of cases) and wild-type (>90% of cases) [14,15]. Diagnosis of TTR-CA requires a high degree of clinical insight: if the patient presents left ventricular (LV) thickening with signs of dyspnea, fatigue, edema or other findings such as apical sparing of LV longitudinal strain on echocardiography and diffuse late-gadolinium enhancement (LGE) on cardiac magnetic resonance (CMR), TTR-CA should be highly suspected. Although these non-invasive imaging modalities can identify the disease, none of them can provide a confirmatory diagnosis. Multi-society guidelines and the expert consensus now recommend bone scintigraphy with ^99m^Tc-PYP, ^99m^Tc-DPD or ^99m^Tc-HMDP as the first-line diagnostic imaging modality in patients with suspected TTR-CA [6,16,17,18]. Although direct comparison data between these bone tracers are still lacking in CA management, there is evidence that ^99m^Tc-DPD/HMDP can also image extracardiac ATTR deposits, while ^99m^Tc-PYP cannot [19].

Cardiac images are interpretated using simple visual scoring and semiquantitative indices. The most widely used visual score is Perugini grade, a comparison of cardiac to bone uptake, giving scores from 0 to 3; grades 2 and 3 are deemed positive for TTR-CA in the absence of clonal dyscrasia, while grade 0 is considered negative for TTR-CA. Grade 1 uptake could represent either AL or TTR-CA [6,20]. Example images of each grade from our institution are depicted in Figure 2. In the largest multicentre study so far involving patients with Perugini grade ≥2 along with negative monoclonal gammopathy, a diagnosis of TTR-CA could be identified without biopsy confirmation (positive predictive and specificity > 98%) [7]. Semiquantitative indices, such as the heart to contralateral thorax (H/CL) ratio and heart/whole body (H/WB) ratio, have been validated for 1 and 3 h imaging and improve the measurement of visual scores [6,21]. Furthermore, a systematic review confirmed that bone scintigraphy using either visual score or semiquantitative indices has greater than 90% sensitivity and specificity for the diagnosis of TTR-CA [8]. Currently, bone scintigraphy has dramatically changed the clinical diagnosis flowchart as an important non-invasive alternative to biopsy for diagnosing TTR-CA [6,7,15,16,18], as shown in Figure 3.

### 3.2. Quantitative SPECT/CT

Compared to planar quantification, single-photon emission tomography (SPECT)/computed tomography (CT) is a hybrid imaging approach consisting of SPECT for functional images and CT for anatomical images. When using quantitative SPECT/CT, more accurate quantitation in achieved using factors such as standardized uptake value (SUV) produced by measurement of injection dose, conversion factors, and counts on images by attenuation correction. SUV measurements in different organs, such as myocardium blood pool and individual bones, can be derived by drawing volumes of interests (VOIs) on SPECT images or CT images and then projecting these onto SPECT images,. Although SPECT is deemed to have less accurate quantification abilities than PET, a meta-analysis revealed that SPECT might be preferable to PET for identifying the subtype of CA [23]. Furthermore, several recent studies [24,25,26] have now expanded on their preliminary works by showing that the quantitative SPECT/CT metrics (SUV_max_, SUV_peak_ and SUV retention index) are practical and have excellent correlations with Perugini scores, but fail to differentiate the myocardial uptake between patients with score 2 and 3. It seems that Perugini scores not only indicate the presence of amyloid deposition but rather a certain amount of amyloid burden, which potentially leads to an underestimation of the disease and false response evaluation. This is because the definition of grade 3 is “mild/absent bone uptake”, which will be confounded by surrounding soft tissue uptake. Therefore, the composite SUV myocardial/soft tissue ratio perhaps offers a means of better quantifying the amyloid burden to distinguish between scores 2 and 3. Further research is required to investigate the role of potential imaging biomarkers which may be validated for reliable response evaluation.

### 3.3. Differentiation between AL and ATTR by SPECT

Previous studies report that the H/CL ratio of ≥1.5 on 1 h images or H/CL ratio ≥ 1.3 on late (3 h) images of ^99m^Tc-PYP scintigraphy can clearly distinguish TTR from AL amyloidosis with over 90% sensitivity and specificity [21,27]. Of note, Perugini grade 1 can be found in as many as 9% of patients with TTR-CA [7], and up to 25% of patients with AL-CA will have grade 2 or 3 radiotracer uptake [28]. Thus, AL amyloidosis should be excluded when ordering a bone scintigraphy scan for diagnosing TTR-CA. On the other hand, in cases with grade 1 in the absence of monoclonal protein, an ongoing EMB or other organ (e.g., salivary glands, abdominal fat pad, skin, or gastrointestinal) biopsy should still be considered to define the presence of CA [28].

### 3.4. Prognostic Value

Prognostic studies of bone scintigraphy are still limited. A visual score of ^99m^Tc-DPD /PYP has not been proven to be an independent predictor of survival in patients with confirmed ATTR [27,29]. Similar results were found in which the H/CL and H/WB ratios were also not associated with outcomes [30,31]. On the contrary, in a multicentre study using ^99m^Tc-PYP scan, a H/CL uptake of 1.6 or greater was associated with worse survival among patients with TTR-CA [27]. These previously contradictory data are difficult to interpret but suggest that H/CL may have a prognostic role in extensive sample analysis. It has been more recently shown that a metric termed the apical sparing ratio, calculated by dividing the apical counts by the sum of base and mid-segment counts on ^99m^Tc-PYP SPECT images, may also predict mortality in patients with TTR-CA [32]. Similarly, a recent Japanese study looking at the prognostic value on semi-quantitative indices of ^99m^Tc-PYP SPECT/CT suggested that the lateral wall-to-cavity counts ratio can independently predict the prognosis of wild-type TTR-CA patients [33]. Nevertheless, because of abnormal comparator uptake or change in comparator uptake over time, the relative nature of these approaches still reduces accuracy in monitoring disease progression or treatment response.

### 3.5. Monitoring Therapy Response

Currently, several pharmacotherapeutic treatments have emerged and have significantly reduced the morbidity and mortality of ATTR. Tafamidis is currently the only drug that has shown efficacy in a randomized trial in patients with both wild-type and variant ATTR. The recommended follow-up scheme for TTR-CA patients involves 6-monthly visits with complete blood tests (NT-proBNP and troponin), ECG, annual echocardiography, and 24 h Holter monitoring [29]. In addition to the conventional evaluation methods, some studies have demonstrated the role of cardiac imaging, including CMR and speckle tracking echocardiography, in monitoring the tafamidis treatment outcomes of TTR-CA patients [34,35]. Although the optimal time frame for monitoring treatment is unknown, we found that the SUV_max_ and SUVR_M,/B_ (SUV_max_ for myocardial to blood pool ratio) both decreased in an 86-year-old man with a diagnosis of TTR-CA following 6 months of tafamidis therapy in our institution, while Perugini grading was unchanged (Figure 4, unpublished data). A similar case report was described by Bellevue D and colleagues [36]. These two cases indicated that quantitative SPECT/CT might be a promising tool for the assessment of therapeutic response and for managing the treatment of TTR-CA. There is also an ongoing clinical trial (REMOD-TTR, NCT04535349) designed to evaluate the effect of 6 months of tafamidis on quantitative and semi-quantitative SPECT/CT assessment of bone-seeking tracers. Further research is needed regarding the choice of appropriate imaging modalities for monitoring therapy-related alterations to best tailor treatment in TTR-CA patients.

## 4. Amyloid-Targeting PET Imaging

### 4.1. Diagnosis

Bone-seeking radiotracers have revolved primarily around TTR-CA diagnosis while offering a low sensitivity for identifying AL-CA. Targeted amyloid PET imaging has the capability to identify all amyloid deposits regardless of the original protein precursor. Multiple radiotracers for amyloid PET have been investigated in Alzheimer’s disease; these tracers are believed to bind with the beta-pleated motif of the amyloid fibril and have shown promise in clinical research studies, as described in the North American and European expert consensus recommendations for cardiac amyloidosis [6,37]. PET imaging using semi-quantitative parameters (target-to-background ratio) or quantitative metrics (SUV or retention index) has been used in several studies and has demonstrated values for the detection of CA [38]. A meta-analysis including ^11^C-PIB, ^18^F-florbetapir, and ^18^F-florbetaben PET studies showed that the combined sensitivity and specificity for the diagnosis of CA was 95% and 98%, respectively [39,40,41]. A recent study using ^18^F-flutemetamol PET/MRI or PET/CT showed it to be positive in only 2 of 12 patients with CA, but there was no difference in SUV_max_ and SUV_mean_ between CA and controls [42]. However, another pilot ^18^F-flutemetamol study demonstrated a higher number of favorable results (8 out of 9 patients with CA) [43]. Due to the contradictory results, further studies with large sample sizes and appropriate imaging protocols are needed in order better to define the accuracy of ^18^F-flutemetamol PET imaging in CA.

### 4.2. Differentiation between AL and ATTR by PET

To date, only small preliminary studies in vivo utilizing these tracers have highlighted potential benefits in stratifying AL-CA from TTR-CA. The cardiac retention index of ^18^F-florbetapir tended to be higher in AL patients but was not significantly less in TTR-CA [38]. Subsequently, a similar finding was reported [41] without being able to separate the two CA subtypes by using dynamic PET with ^18^F-florbetaben in a 20 min scan. These initial findings were also re-observed on ^11^C-PiB PET imaging [40,44,45]. More recently, a dual-center study demonstrated that ^11^C-PiB PET imaging had 100% diagnostic accuracy of visual assessment in AL amyloidosis. Moreover, the myocardium to blood SUV ratio was significantly higher in AL than in ATTR patients [46]. However, since ^11^C-PiB binding to cardiac amyloids has a larger variability in retention index [47], and an on-site cyclotron is required for the production of ^11^C-labeled radiotracer, ^18^F-labeled PET tracers with a longer half-life (110 min) may overcome these drawbacks and may be more widely used.

^18^F-florbetaben PET imaging with a dynamic and delayed acquisition shows a significant increase in uptake in patients with AL compared to those with ATTR or without CA. However, in early studies, there was no difference between AL and TTR amyloidosis patients [48]. In addition, there was virtually no difference in the retention index between TTR and controls. These observations strongly suggested that a late scan obtained at least 30 min after ^18^F-florbetaben injection can reliably discriminate CA due to AL from either TTR or other mimicking conditions (Figure 5 [48]). Notably, because of the non-specific affinity to TTR-CA patients, the application of ^18^F-florbetaben and PET imaging for the identification of TTR in patients with suspected CA may not be recommended. Furthermore, a dynamic ^18^F-florbetaben PET study demonstrated that using kinetic model-fitting parameters allows for distinguishing between the two types of amyloid pathology and strengthens the diagnostic accuracy for AL-CA [49]. With the development and application of Deep Learning (DL) techniques in the field of medical imaging, it was reported that a simple DL model derived from cardiac ^18^F-florbetaben PET images acquired a few minutes after the injection could be used to help clinicians differentiate AL from TTR-CA [50]. Therefore, although the clinical manifestations of AL and ATTR amyloidosis overlap, ^18^F-florbetaben PET will likely open new avenues as the preferred imaging modality in patients with suspected AL amyloidosis.

### 4.3. Prognostic Value

Conventional serum biomarkers such as NT-proBNP and troponin have demonstrated prognostic value in patients with CA, as they indicate cardiac involvement of systemic amyloidosis. However, only a few studies have investigated the role of amyloid-targeting PET in this specific patient group. Recently, Lee SP et al. [40] reported that quantification of myocardial ^11^C-PiB uptake by PET was associated with adverse outcomes independently among patients with AL-CA. Interestingly, the degree of myocardial ^11^C-PiB uptake seems to be a better predictor for clinical events such as all-cause death or acute decompensated heart failure than LVEF or diastolic filling parameters (E/e’ ratio) assessed by echocardiography. Furthermore, Choi YJ et al. [51] showed that ^11^C-PiB PET/CT was not only a strong independent predictor of one year overall survival, but also provided incremental prognostic benefits when combined with commonly used serum biomarkers such as troponin I, NT-proBNP, and dFLC in patients with AL-CA. Given the need for improved cardiac staging systems for prognostication in AL, future prospective studies, possibly multicenter studies, are warranted to discover whether PET/CT should be incorporated into risk stratification for AL amyloidosis patients.

### 4.4. Therapy Response Evaluation

Therapy response for AL amyloidosis is normally assessed by hematology (decreased light chains affected and normalization of bone marrow) and organ involvement. For CA, reduction in NT-proBNP by >30% and 300 ng/L (if baseline NT-proBNP ≥ 650 ng/L) is defined as a cardiac response [52]. However, there are drawbacks of solely using natriuretic peptides when they are influenced by other factors (e.g., renal dysfunction, atrial fibrillation and lower body mass index) which are common comorbidities in patients with AL-CA [53]. Thus, imaging biomarkers that reflect a cardiac response could be very useful in detecting changes in structure or function associated with biomarker variation following treatment.

A non-invasive method for detecting and quantifying cardiac amyloid depositions with ^18^F-florbetapir was evaluated in patients with AL-CA prior to and post-chemotherapy, and its serial utility in monitoring was also assessed [54]. In the three patients with complete hematological response, although no significant difference in cardiac uptake was found between the first and repeated images, there was a suggestion that treatment-naïve patients may have higher cardiac uptake. Given the prospective nature of this study, future studies are needed to include more significant numbers of patients and to assess the optimal time of repeat ^18^F-florbetapir imaging after treatment. Furthermore, another cardiac ^18^F-florbetapir PET study consistently showed that all AL patients in that cohort had demonstrable cardiac uptake, despite hematologic remission for more than 1 year [55]. The results offer new insight into the argument of “active” vs. “inactive” amyloid hearts [46], irrespective of serum FLC burden response. In addition to ^18^F-florbetapir, the use of ^18^F-florbetaben PET/CT for monitoring anti-inflammatory (AA), anti-myeloma (AL) and TTR-stabilizing (TTR) therapy were also evaluated [56]. The results showed that changes in myocardial tracer retention from baseline to follow-up corresponded well with treatment response.

In summary, the theoretical benefits of amyloid PET in assessing prognosis and monitoring treatment response allow more widespread use in CA. Furthermore, this method will shed much light on a currently opaque and ambiguous assessment process. Nevertheless, it is not yet possible to make reliable statements regarding the incorporation of amyloid PET into the CA-imaging algorithm due to limited data (mostly retrospective) [57], and there are no clinically available PET tracers authorized in the European Union or the United States. Currently, a larger prospective phase III trial evaluating the utility of ^18^F-florbetaben PET as a diagnostic method for AL-CA is underway (NCT05184088).

## 5. Future Perspective and Conclusions

With the development of non-invasive diagnostic approaches and the emergence of new pharmacotherapeutic treatments for CA, new areas and unsolved questions have arisen. Despite the transformative effects of bone scintigraphy being readily accessible to diagnose ATTR-CA, global experiences regarding serial imaging with bone scintigraphy for the evaluation of prognosis and disease progression or improvement of disease state are still relatively scarce. Further larger clinical trials are needed to determine whether quantitative SPECT/CT measurements provide prognostic information or contribute to assess disease progression and treatment for TTR-CA.

PET imaging is another promising, non-invasive option for the diagnosis of CA and may help differentiate between ATTR and AL amyloidosis. On the one hand, there are an increasing number of new PET tracers for the diagnosis of CA, such as targeting activated fibroblasts probe ^68^Gallium-labeled fibroblast activation protein inhibitor (^68^Ga-FAPi), which has been reported to have a high uptake in the thickened left ventricle myocardium and tongue in patients with AL amyloidosis [58]. Another bone-avid tracer, ^18^F-NaF, has shown weak visual uptake and low target-to-background ratio inferior to that of ^99m^Tc-PYP SPECT for the diagnosis of TTR-CA [59,60,61]. On the other hand, a simultaneous PET/MRI system may be useful for a one-stop-shop assessment of amyloid deposition, tissue characterization, and cardiac function in patients needing rapid diagnosis of CA [60,62,63], which will pave the way for new comprehensive applications and characterizations.

Finally, with targeted therapies for TTR-CA and life-saving treatments for AL-CA already available, this emerging field needs a quantitative measurement of cardiac amyloid burden. Bone-seeking SPECT/CT quantification and amyloid-targeting PET imaging could be used for monitoring treatment to provide more insights into treatment response assessed by quantifying amyloid burden in the future.

## Figures and Tables

**Figure 1 biology-11-01395-f001:**
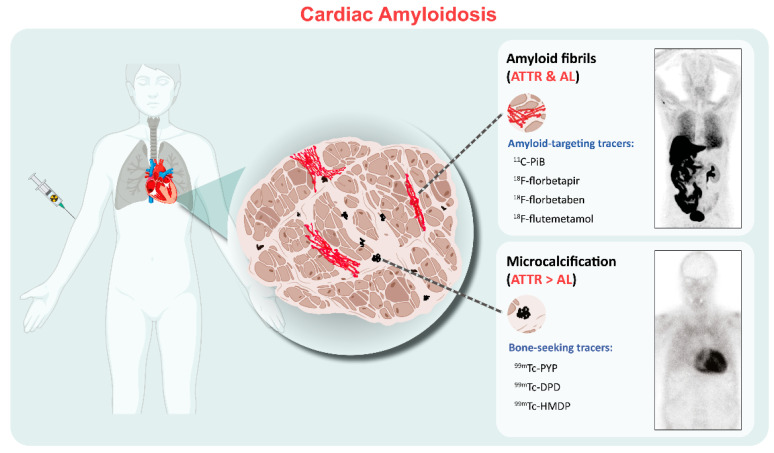
Binding properties of the clinically used radiotracers for cardiac amyloidosis. The amyloid-targeting PET tracers including ^11^C-PiB and ^18^F-florbetapir/florbetaben/flutemetamol bind to both AL and ATTR amyloid fibrils. SPECT bone-seeking tracers, including ^99m^Tc-PYP/DPD/HMDP, show more abundant uptake in ATTR than those in AL, possibly related to fibril microcalcification. Examples include ^11^C-PIB PET/CT and ^99m^Tc-DPD scintigraphy images. PET= positron emission tomography; ATTR = transthyretin amyloidosis; AL = light-chain amyloidosis.

**Figure 2 biology-11-01395-f002:**
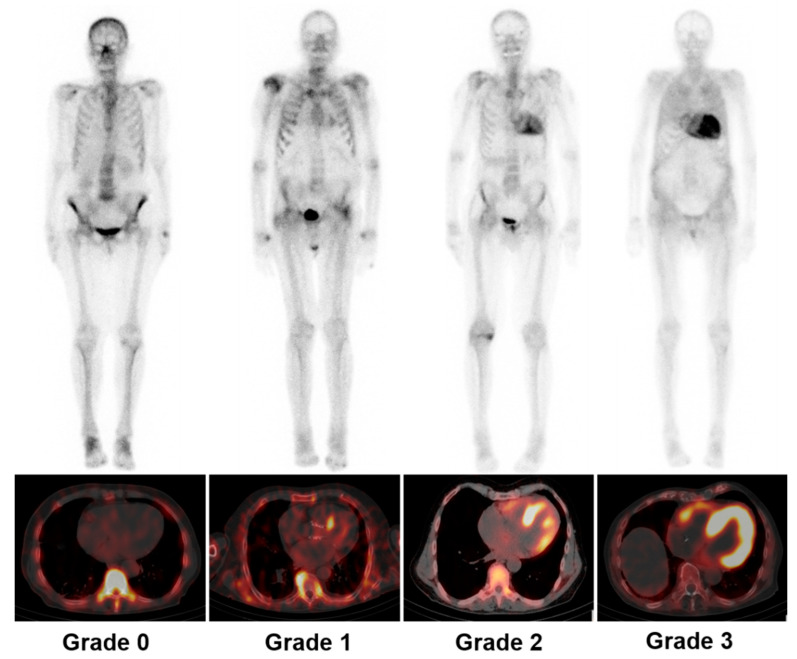
Representative examples of Perugini grading scale for cardiac uptake by using ^99m^Tc-DPD whole-body scintigraphy (**upper panel**) and SPECT/CT (**lower panel**). Grade 0: unaffected individual without any cardiac tracer uptake; grade 1: patient with AL and mild cardiac uptake; grade 2: patient with ATTR and strong cardiac uptake greater than bone uptake; grade 3: patient with ATTR and pronounced tracer uptake in the myocardium but reduced bone uptake. SPECT/CT = single-photon emission tomography/computed tomography; AL = light-chain amyloidosis; ATTR = transthyretin amyloidosis.

**Figure 3 biology-11-01395-f003:**
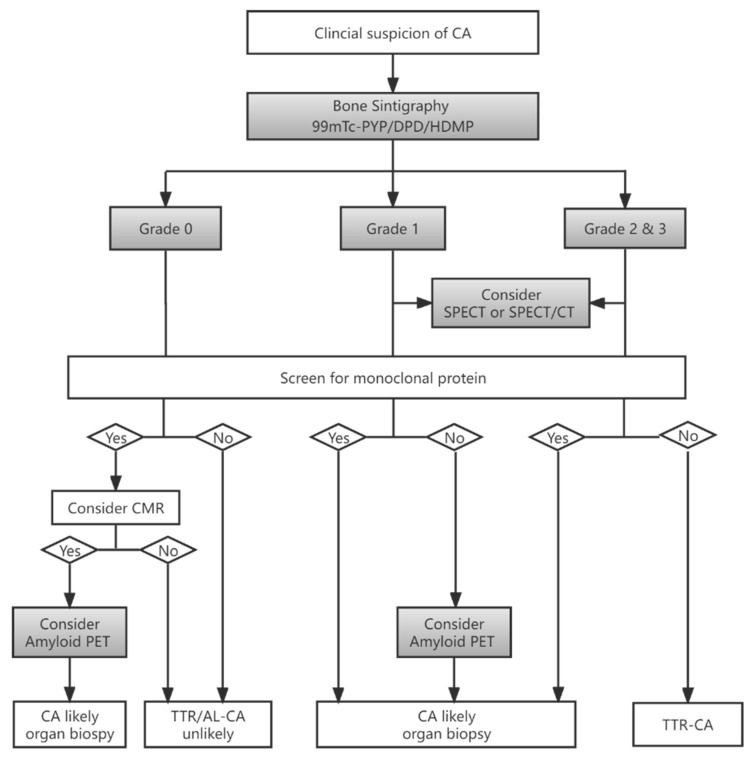
Flowchart for diagnosis of CA based on bone scintigraphy in patients with suspected CA (adapted data from [18,22]). CA = cardiac amyloidosis; TTR = transthyretin, AL = light-chain amyloidosis; SPECT/CT = single-photon emission tomography/computed tomography; PET = positron emission tomography; CMR = cardiac magnetic resonance.

**Figure 4 biology-11-01395-f004:**
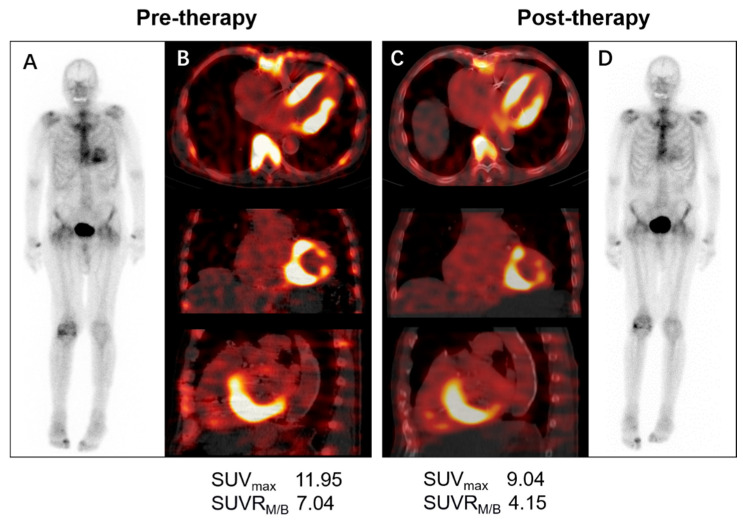
Serial ^99m^Tc-DPD whole-body planar and SPECT/CT scans in a patient with wild-type ATTR receiving 6 months of 61 mg tafamidis. The visual cardiac uptake (Perugini grade 2) was unchanged before (**A**) and after tafamidis therapy (**D**), whereas quantitative measurement of left ventricular uptake on axial, coronal, and sagittal SPECT/CT images before (**B**) and after tafamidis (**C**) showed a decrease in SUV_max_ (11.95 vs. 9.04 g/mL) and SUVR_M/B_ (7.04 vs. 4.15). SPECT/CT=single-photon emission tomography/computed tomography; ATTR = transthyretin amyloidosis; SUV_max_ = maximum standardized uptake value; SUVR_M/B_ = SUV_max_ for myocardial to background (blood pool) ratio.

**Figure 5 biology-11-01395-f005:**
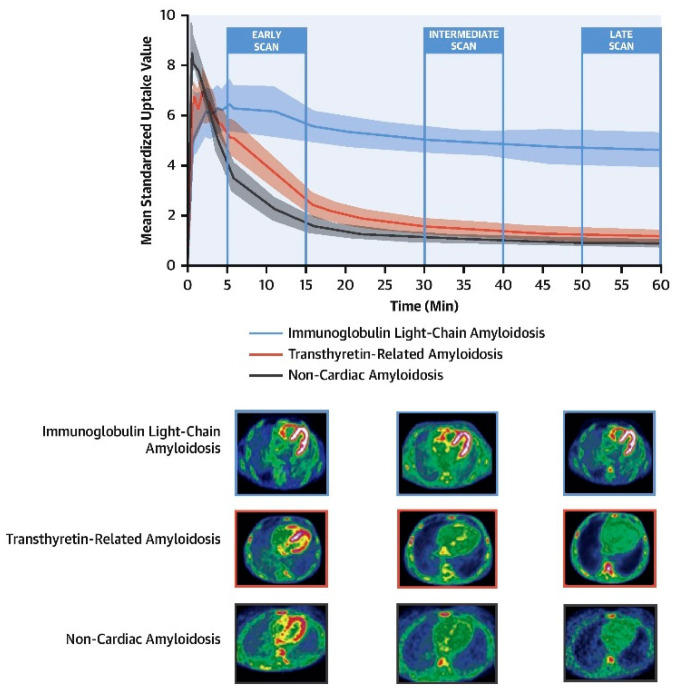
Cardiac ^18^F-florbetaben PET scan in patients with and without CA. (**Upper panel**) The myocardial time–activity curves derived from dynamic ^18^F-florbetaben PET scans in patients with non-CA (black), ATTR-CA (red), and AL-CA (blue). A shaded area for each curve represents the 95% confidence interval. (**Lower panel**) Early (5 to 15 min), intermediate (30 to 40 min), and late (50 to 60 min) ^18^F-florbetaben cardiac PET scans in patients with AL-CA, ATTR-CA and non-CA. CA = cardiac amyloidosis; AL = light-chain amyloidosis; ATTR = transthyretin amyloidosis; PET = positron emission tomography;. Reprinted from [48], copyright (2021), with permission from Elsevier.

## Data Availability

Not applicable.

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
