# Peer review of "Nuclear Molecular Imaging of Disease Burden and Response to Treatment for Cardiac Amyloidosis"

_biology, 2022, doi:10.3390/biology11101395_

Round 1

Reviewer 1 Report

Nuclear molecular imaging of disease burden and response to treatment for cardiac amyloidosis is a well written and complete review of molecular imaging of cardiac amyloidosis.  There are only a few suggestions and corrections.

AA needs to be defined, line 35

It would be useful to the reader to suggest a routine clinical imaging algorithm/protocol to evaluate for cardiac amyloidosis.  Specifically, suggest if both planar and SPECT imaging be performed and at which time points SPECT should be performed.  Several metrics have been noted (H/CL and H/WB) but there is not guidance for the reader as to which should be routinely performed.  Also, it would be helpful to the reader to know when and if to incorporate PET agents in the imaging algorithm. 

Please comment on the nuances between PYP and HDP interpretation for routine clinical care.

Figure 2.  May be useful to demarcate visually that grade 2/3 are considered positive for ATTR and grade 1/2 is negative.

May be of benefit to state where PET studies are being performed as a clinical standard of care for cardiac amyloidosis.  In the United States, PET agents are not currently reimbursed.

Author Response

1.AA needs to be defined, line 35

Response:we thank the Reviewer for pointing this out – we have added the full name of AA in Line 41 of the revised manuscript.

2.It would be useful to the reader to suggest a routine clinical imaging algorithm/protocol to evaluate for cardiac amyloidosis.  Specifically, suggest if both planar and SPECT imaging be performed and at which time points SPECT should be performed.  Several metrics have been noted (H/CL and H/WB) but there is not guidance for the reader as to which should be routinely performed.  Also, it would be helpful to the reader to know when and if to incorporate PET agents in the imaging algorithm.

Response: we appreciate the Reviewer’s suggestion. We added a diagnostic algorithm for patients with suspected cardiac amyloidosis as shown in Fig 3 in the revised manuscript.

Additionally, in clinical practice for PYP scan, both visual scoring and H/CL ratio are recommended by

ASNC/AHA/ASE/EANM/HFSA/ISA/SCMR/SNMMI expert consensus for CA[1,2]. Diagnosis of ATTR cardiac amyloidosis cannot be made solely based on H/CL ratio alone with PYP. H/CL ratio is not recommended if there is absence of myocardial uptake on SPECT. Additionally, if the visual grade is 2 or 3, diagnosis is confirmed and H/CL ratio assessment is not necessary. H/CL ratio is typically concordant with visual grade. If discordant or the visual grade is equivocal, H/CL ratio may be helpful to classify equivocal visual grade 1 vs 2 as positive or negative[2]. Only PYP/DPD/HMDP visual scoring has been used for clinical diagnosis criteria for ATTR cardiac amyloidosis[1]. H/WB ratio has been investigated in some DPD/HMDP studies, but it was not recommended by any guideline or expert consensus as a routine measurement in clinical practice.

3.Please comment on the nuances between PYP and HDP interpretation for routine clinical care.

Response: HMDP is sometimes abbreviated HDP. To the best of our knowledge, HDMP has been approved for CA in most European countries, but not in Unite State. Current expert consensus recommends that all 99mTc phosphate derivatives including DPD, PYP, and HMDP are suitable for evaluating ATTR-CA. However, direct comparison study between HMDP and other bone tracer is lacking in CA management.

Interpretation of HMDP scans for CA is very analogous to PYP with some minor differences: As opposed to PYP, use of the H/CL ratio is not recommended for HMDP due to background noise of soft tissue(skeletal muscle and lung) uptake[3]. Moreover, HMDP has been shown to have a greater ability to identify systemic amyloidosis than other bone tracers, so extracardiac uptake is also recommended to report in HMDP imaging[4].

4.Figure 2. May be useful to demarcate visually that grade 2/3 are considered positive for ATTR and grade 1/2 is negative.

Response: While we appreciate the reviewer’s feedback, we do not fully agree. Grade 2 or 3 uptake is consistent with ATTR in the absence of clonal dyscrasia, while grade 0 is considered negative for TTR-CA, grade 1 uptake could represent either AL or TTR-CA. We have added the suggested content in Line 128-130 of revised manuscript.

5.May be of benefit to state where PET studies are being performed as a clinical standard of care for cardiac amyloidosis.  In the United States, PET agents are not currently reimbursed.

Response: Thank you for Reviewer's comment. Although amyloid-targeting PET is a promising imaging modality for CA, as far as we know, there is no clinically available PET tracers that was authorized in EU and Unite State. Recently, a larger phase III clinical trial evaluating the utility of 18F-florbetaben PET as a diagnostic tool for AL-CA is underway(NCT05184088). We added this content in Line 362-367 of revised manuscript.

Reference:

1.Dorbala, S.; Ando, Y.; Bokhari, S.; Dispenzieri, A.; Falk, R.H.; Ferrari, V.A.; Fontana, M.; Gheysens, O.; Gillmore, J.D.; Glaudemans, A.; et al. ASNC/AHA/ASE/EANM/HFSA/ISA/SCMR/SNMMI expert consensus recommendations for multimodality imaging in cardiac amyloidosis: Part 2 of 2-Diagnostic criteria and appropriate utilization. J Nucl Cardiol 2020, 27, 659-673, doi:10.1007/s12350-019-01761-5.

2.Dorbala, S.; Ando, Y.; Bokhari, S.; Dispenzieri, A.; Falk, R.H.; Ferrari, V.A.; Fontana, M.; Gheysens, O.; Gillmore, J.D.; Glaudemans, A.; et al. ASNC/AHA/ASE/EANM/HFSA/ISA/SCMR/SNMMI expert consensus recommendations for multimodality imaging in cardiac amyloidosis: Part 1 of 2-evidence base and standardized methods of imaging. J Nucl Cardiol 2019, 26, 2065-2123, doi:10.1007/s12350-019-01760-6.

3.Miller, E.J.; Campisi, R.; Shah, N.R.; McMahon, S.; Cuddy, S.; Gallegos-Kattan, C.; Maurer, M.S.; Damy, T.; Slart, R.; Bhatia, K.; et al. Radiopharmaceutical supply disruptions and the use of (99m)Tc-hydroxymethylene diphosphonate as an alternative to (99m)Tc-pyrophosphate for the diagnosis of transthyretin cardiac amyloidosis: An ASNC Information Statement. J Nucl Cardiol 2022, doi:10.1007/s12350-022-03059-5.

4.Rapezzi, C.; Gagliardi, C.; Milandri, A. Analogies and disparities among scintigraphic bone tracers in the diagnosis of cardiac and non-cardiac ATTR amyloidosis. J Nucl Cardiol 2019, 26, 1638-1641, doi:10.1007/s12350-018-1235-6.

Reviewer 2 Report

I would like to thank the authors for this very well written and interesting review article. While amyloid imaging is regularly clinically used for brain assessments, it is rather new to cardiac imaging. Here it opens a lot of new opportunities. The authors compile and present very well the current state of science and identify gaps that need to be further studied to bring it close to clinical routine use. The review is very balanced in that way and I am looking forward to read more about the development of the field of cardiac amyloid imaging in the future.

Author Response

we really appreciate the Reviewer’s comment.

Reviewer 3 Report

In this manuscript, authors reviewed and discussed the current understanding of bone-seeking scintigraphy and amyloid-targeting PET for the diagnosis, prognosis, and monitoring treatment in cardiac amyloidosis.

1. There are some grammar mistakes in the manuscript, such as line 48-49, “to ameliorate TTR amyloidosis associated with outcomes” should be “TTR amyloidosis associated outcomes”, Line 100-101, “none of them still couldn’t address a confirmatory diagnosis”, assuming authors mean “none of them still could address”? Such errors exist in this manuscript, please double check and correct all.

2. Please make sure that all statements are properly cited. There are missing references for cited data from previous studies.

3. For all figures of representative examples, please clarify whether those are from previous studies or authors’ own data?

4. For section 3.2 and 3.3, please first discuss SPECT/CT before differentiation between AL and ATTR by SPECT for a better logic flow.

5. Line 182-185, authors discussed their own findings, are those data published or unpublished? Please properly cite previous findings or clarify unpublished data.

Author Response

1.There are some grammar mistakes in the manuscript, such as line 48-49, “to ameliorate TTR amyloidosis associated with outcomes” should be “TTR amyloidosis associated outcomes”, Line 100-101, “none of them still couldn’t address a confirmatory diagnosis”, assuming authors mean “none of them still could address”? Such errors exist in this manuscript, please double check and correct all.

Response:Thank you for pointing this out – we have corrected this in the revised manuscript.

2.Please make sure that all statements are properly cited. There are missing references for cited data from previous studies.

Response:Thank you for pointing this out – we have cited the missing references in the revised manuscript.

3 .For all figures of representative examples, please clarify whether those are from previous studies or authors’ own data?

Response:Thank you for pointing this out –– we have clarified in Line 128-129 of the revised manuscript.

4. For section 3.2 and 3.3, please first discuss SPECT/CT before differentiation between AL and ATTR by SPECT for a better logic flow.

Response: Thank you for this suggestion – we have switched the order of section 3.2 and 3.3.

5. Line 182-185, authors discussed their own findings, are those data published or unpublished? Please properly cite previous findings or clarify unpublished data.

Response:Thank you for pointing this out. The figure 3 we discussed in line 182-185 was unpublished data from our institution, we have clarified in Line 233 of the revised manuscript.

.